# Effects of Experimental Nitrogen Addition on Nutrients and Nonstructural Carbohydrates of Dominant Understory Plants in a Chinese Fir Plantation

**Fangchao Wang [1], Fusheng Chen [1,2,*], G. Geoff Wang [3], Rong Mao [1,2] [ID], Xiangmin Fang [1], Huimin Wang [4] and Wensheng Bu [1,2]**

1   Jiangxi Provincial Key Laboratory of Silviculture, Jiangxi Agricultural University, Nanchang 330045, China; wfangch@163.com (F.W.); maorong23@163.com (R.M.); xmin007@163.com (X.F.); bws2007@163.com (W.B.)
2   Jiulianshan National Observation and Research Station of Chinese Forest Ecosystem, Key Laboratory of National Forestry and Grassland Administration on Forest Ecosystem Protection and Restoration of Poyang Lake Watershed, Jiangxi Agricultural University, Nanchang 330045, China
3   Department Forestry and Environmental Conservation, Clemson University, Clemson, SC 29634, USA; gwang@g.clemson.edu
4   Qianyanzhou Ecological Station, Key Laboratory of Ecosystem Network Observation and Modeling, Institute of Geographic Sciences and Natural Resources Research, Chinese Academy of Sciences, Beijing 100101, China; wanghm@igsnrr.ac.cn
*   Correspondence: chenfusheng@jxau.edu.cn; Tel.: +86-791-838-13243

**Abstract:** *Research Highlights:* This study identifies the nitrogen (N) deposition effect on understory plants by altering directly soil nutrients or indirectly altering environmental factors in subtropical plantation. *Background and Objectives:* N deposition is a major environmental issue and has altered forest ecosystem components and their functions. The response of understory vegetation to N deposition is often neglected due to a small proportion of stand productivity. However, compared to overstory trees, understory species usually have a higher nutrient cycle rate and are more sensitive to environmental change, so should be of greater concern. *Materials and Methods:* The changes in plant biomass, N, phosphorus (P), and nonstructural carbohydrates (NSCs) of three dominant understory species, namely *Dicranopteris dichotoma*, *Lophatherum gracile*, and *Melastoma dodecandrum*, were determined following four years of experimental N addition (100 kg hm$^{-2}$ year$^{-1}$ of N) in a Chinese fir plantation. *Results:* N addition increased the tissue N concentrations of all the understory plants by increasing soil mineral N, while N addition decreased the aboveground biomass of *D. dichotoma* and *L. gracile* significantly—by 82.1% and 67.2%, respectively. The biomass of *M. dodecandrum* did not respond to N addition. In contrast, N addition significantly increased the average girth growth rates and litterfall productivity of overstory trees—by 18.28% and 36.71%, respectively. NSCs, especially soluble sugar, representing immediate products of photosynthesis and main energy sources for plant growth, decreased after N addition in two of the three species. The plant NSC/N and NSC/P ratios showed decreasing tendencies, but the N/P ratio in aboveground tissue did not change with N addition. *Conclusions:* N addition might inhibit the growth of understory plants by decreasing the nonstructural carbohydrates and light availability indirectly rather than by changing nutrients and N/P stoichiometry directly, although species-specific responses to N deposition occurred in the Chinese fir plantation.

**Keywords:** experimental nitrogen addition; understory plant growth; plant nutrient; nonstructural carbohydrates

## 1. Introduction

Fossil fuel combustion and chemical fertilization are increasing atmospheric nitrogen (N) deposition throughout the world and altering regional and global N cycles [1]. The annual input of reactive N into the Earth's land surface has approximately doubled since 1970, and this trend will continue in the rapidly developing regions of the world [2]. Considering that N is an essential nutrient element limiting plant photosynthetic capacity and productivity, increased N deposition would produce a cascading effect on forest ecosystem structure, process, and function [3–5]. The understory plants' responses to N deposition in forest ecosystems need to be of greater concern in the future.

The vegetation of forest ecosystems includes both overstory (or canopy) and understory species. Because of the dominance of the forest canopy, understory species and their responses to environmental changes are often neglected, especially in plantation forests [6]. Understory vegetation, despite often accounting for a small proportion of stand productivity, is an important component of forest ecosystems and plays a key role in regulating ecosystem processes and functions. Moreover, compared with tree species in the canopy, understory species usually have faster nutrient turnover rates and thus are more sensitive to environmental change. Previous studies have found that N deposition exerted a substantial influence on soil nutrient supply and decreased understory plant biomass due to increasing soil acidification and phosphorus (P) limitation [7–9]. Furthermore, some studies reported that increased N availability exaggerated asymmetrical competition for other resources (e.g., light and water), favoring the growth of overstory species at the expense of understory species [6,10,11]. For example, Strengbom suggested that understory vegetation was mainly limited by light, since N addition increased shading from the canopy that decreased the light available to understory plants in a boreal forest [12]. However, these studies concentrated on boreal and temperate natural forests, the results from which may not be applicable to subtropical plantations given that species coexistence and ecosystem biogeochemical cycles vary across forest types and climatic regions [6,13,14].

Nitrogen and P are often recognized as limiting nutrients in forest ecosystems; thus, N and P concentrations and their stoichiometry have been widely used to determine plant adaptation and feedback in response to resource alteration [15]. Nonstructural carbohydrates (NSCs) are the immediate products of photosynthesis and are often used to indicate plant light environment and growth rate [16]. The proportional relationships among N, P, and NSCs to a large extent reflect the available C and energy utilization efficiency of plant growth [17]. For example, N deposition increases N supply and causes an imbalance between N and P in plant tissues [18]. Furthermore, N deposition alters plant photosynthetic processes and the associated accumulation and consumption of NSCs including starch (ST) and soluble sugar (SS) [19]. Therefore, identifying NSCs and their interactions with N and P in aboveground (i.e., leaves) and belowground tissues (i.e., roots) might provide a new perspective on the effect of N deposition on understory vegetation growth.

In this study, a chronic N addition experiment was conducted to simulate N deposition in a Chinese fir (*Cunninghamia lanceolata*) plantation of subtropical China. We measured soil available N, P and plant biomass and N, P, and NSC concentrations in major understory plants to assess the responses of understory vegetation to N deposition in subtropical plantations. We hypothesized that (1) N deposition leads to an imbalance between N and P in soils and understory plants due to elevated N availability; (2) N deposition causes a decline in understory plant biomass due to elevated P limitation or increased shading by stimulated overstory growth; and (3) the changes in NSCs and NSCs/nutrients in aboveground and belowground tissues help identify the potential resource (such as light) competition mechanisms of understory plants in response to N deposition [14]. Our results may have some implications for understory plant management in plantation forests of subtropical areas experiencing N deposition.

## 2. Materials and Methods

### 2.1. Study Region

This study was conducted in a 12-year-old Chinese fir planation in the Jian-Taihe Basin of Jiangxi Province, China ($26°42'$ N, $115°04'$ E, 100 m asl). The area is a typical red soil hilly region with a subtropical moist monsoon climate. The soil belongs to the typical Hapludult Ultisols with 68% sand and 15% clay [20]. The month average temperatures range from 6.5 °C in January to 29.7 °C in July, and a mean annual temperature is 18 °C. The annual precipitation ranges from 945 to 2144 mm, with an average of 1500 mm [21]. The area belongs to the core distribution of Chinese fir and the center of N deposition in China with 49 kg N ha$^{-1}$ year$^{-1}$ [3,22].

### 2.2. Experimental Treatments and Sample Collection

The simulated N deposition experiment followed a paired design and was established in November 2011. Within each paired plot, two 20 m × 20 m plots were treated with four years of in situ N addition (100 kg hm$^{-2}$ year$^{-1}$ of N) or no N addition (control, CK), with a buffer zone of more than 20 m between the plots. Four replicates were established on four separate hilly slopes. Nitrogen mixed with sand was added four times each year (March, June, September, and December) in the form of NH$_4$NO$_3$. In order to evenly spread NH$_4$NO$_3$ in the N addition plots, we added N together with a small amount of sand (8 kg plot$^{-1}$) and also added sand in control plots. For fertilization we generally chose a date without rain in the two days before or two days after, combined with a local weather forecast. General properties in the Chinese fir forest plantation before the experiment were not significantly different between the experimental units that received the two treatments (Table A1).

In August 2015, eight 1 m × 1 m sample subplots were randomly established in each plot. The understory species in these subplots were measured and recorded to obtain their richness. We used the number of each understory species in subplot as understory richness. Furthermore, the harvested understory plants were divided into three representative species, namely *Dicranopteris dichotoma* (a fern belonging to Gleicheniaceae; a sun plant), *Lophatherum gracile* (a perennial grass belonging to Gramineae; a neutral plant), and *Melastoma dodecandrum* (a creeping small shrub belonging to Melastomataceae; a shade plant) (see Figure A1), and other understory plants. The average proportion of biomass contributed by the three major plants to the total understory vegetation biomass in our study plots was more than 90%. All three species are perennial understory plants. We collected samples of all three understory species in maturation stage. The aboveground tissues (leaves and stems) of these species in each subplot were brought back to the laboratory to measure their dry biomass. We did not harvest the belowground tissue (roots) in order to avoid damage to these permanent study plots.

In addition, we collected samples of aboveground and belowground tissues and from rhizosphere and bulk soils for each of the three understory species within a plot. Soil strongly adhering within 4 mm of roots was considered rhizosphere soil, and the remaining soil was considered bulk soil [23]. Rhizosphere soil samples were collected for each species by separating soil from roots through hand shaking, while the bulk soil samples were collected using a soil auger under/near the plant crown. Each soil sample was divided into two replicates: one used for available nutrient measurement within five days and another stored in a refrigerator at 4 °C and then used for determination of soil enzyme activities. All the plant samples were immediately microwaved for 90 s to stop all enzymatic activity [24], washed with distilled water and oven-dried to a constant mass at 60 °C. After being finely ground using a mixer mill and through a sieve (<0.2 mm), these plant samples were used for determining nutrient and NSC concentrations.

### 2.3. Soil Nutrient and Enzyme Measurement

Soil NH$_4^+$-N and NO$_3^-$-N were extracted with 2 M KCl for 30 min and then measured by spectrophotometry using the indophenol blue and cadmium reduction methods, respectively. Soil available P was extracted with 0.5 M NaHCO$_3$ for 30 min and determined using the

molybdenum-antimony colorimetric method [25]. The activities of N-acetyl-β-D-glucosaminidase (NAG; EC 3.2.1.14) and acid phosphatase (AP; EC 3.1.3.2) in soils were determined by the fluorogenic microplate method [26].

### 2.4. Understory Plant Nutrient Measurement

Plant N and P concentrations were determined by the Kjeldahl method and the molybdenum blue spectrophotometric procedure, respectively, after the samples were digested with $H_2SO_4$ [25]. NSCs were measured by the anthrone colorimetry method [27]. Briefly, the powdered plant sample (0.5 g) was put into a 15 mL centrifuge tube, where 10 mL of 80% alcohol was added. The mixture was incubated in a 100 °C water bath for 20 min and then centrifuged at 4000 rpm for 10 min. The supernatants were retained for SS determination, and the residue was extracted two more times as described above. ST was extracted from the ethanol-insoluble pellet until ethanol was first removed by evaporation. The residue remaining after the SS extraction was extracted with 5 mL 1 M $H_2SO_4$, and the mixture was shaken for 15 min. The mixture was incubated in an 80 °C water bath for 40 min and then centrifuged at 4000 rpm for 10 min. The pellets were extracted two more times with 1 M $H_2SO_4$. SS and ST determinations were performed based on absorbance at 625 nm using the same anthrone reagent in a spectrophotometer [28,29]. NSC concentration was obtained by the sum of the total SS and ST.

### 2.5. Overstory Tree Growth and Litterfall Production Measurement

In each plot, 20 trees have been randomly selected to measure the girth growth rate at breast height using a self-made dendrometer (including a sheet steel, a wire spring two steel nails and a digital caliper) [3] in June and December since the establishment of the experiment. Meanwhile, five 75 cm × 75 cm litter traps were uniformly distributed under the stand canopies to measure litterfall biomass from November 2015 to April 2016. The growth rate and litterfall productivity of Chinese fir were calculated at a plot level in this study in order to assess the potential effect of overstory trees on light conditions of understory plants.

### 2.6. Data Analysis

The data were tested for homogeneity of variances (Brown and Forsythe's variation of Levene's test) before statistical analysis. A paired t-test was used to compare the differences in soil and plant variables between the control and N addition treatment. Multi-way analysis of variance (ANOVA) was used to determine the interactive effects of N addition, plant species and plant tissue on plant nutrient and NSC concentrations. Pearson's correlation analysis was performed to determine the relationship among soil available nutrients, plant nutrients and NSC parameters. All ANOVA and correlation analyses were conducted with a significance criterion of $p < 0.05$ using IBM SPSS 19 statistical software (SPSS, Chicago, IL, USA).

## 3. Results

### 3.1. Plant Growth and Soil Nutrients

After 4 years of N addition, the aboveground biomass of *D. dichotoma* and *L. gracile* significantly decreased by 82.1% and 67.2%, respectively ($p < 0.05$), while the biomass of *M. dodecandrum*, which was much lower than that of the other two species, did not respond to N addition (Table 1). Moreover, N addition did not alter the richness of understory plants (Table 1). In contrast, the average girth growth rates of overstory trees (Chinese fir) within four years after N addition treatment and litterfall productivity in the fifth year significantly increased by 18.28% and 36.71%, respectively (Figure 1, Table A2).

**Table 1.** The aboveground biomass and richness of the major understory plants in the Chinese fir forest plantation treated by nitrogen addition or in the control in 2015.

| Variables | Control | Nitrogen Addition | *T*-test |
|---|---|---|---|
| *Dicranopteris dichotoma* | | | |
| Biomass (kg ha$^{-1}$) | 652.51 ± 83.71 | 117.37 ± 38.49 | $p < 0.05$ |
| Richness | 4.00 ± 1.01 | 2.25 ± 0.63 | ns |
| *Lophatherum gracile* | | | |
| Biomass (kg ha$^{-1}$) | 205.73 ± 16.15 | 67.84 ± 33.96 | $p < 0.05$ |
| Richness | 7.67 ± 1.15 | 4.00 ± 2.45 | ns |
| *Melastoma dodecandrum* | | | |
| Biomass (kg ha$^{-1}$) | 34.33 ± 16.40 | 14.80 ± 8.22 | ns |
| Richness | 3.57 ± 0.81 | 3.67 ± 1.20 | ns |

The values are the means ± SE (*n* = 4). ns = not significant at $p > 0.05$ level.

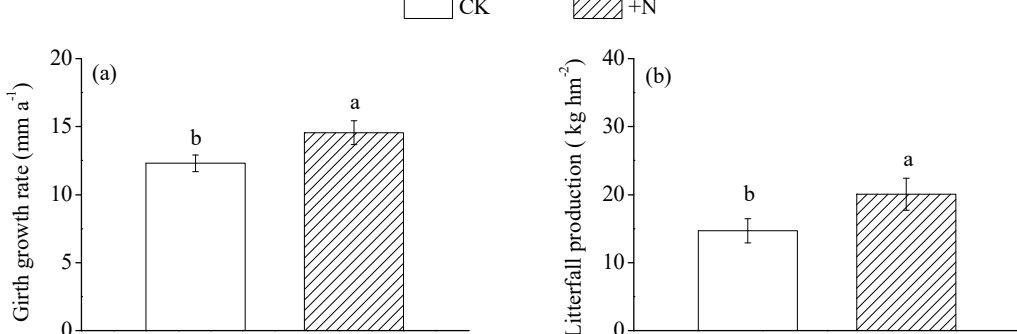

**Figure 1.** Girth growth from 2011 to 2015 (**a**) and litterfall production in 2016 (**b**) of the Chinese fir tree plantation treated by nitrogen addition (+N) or in the control (CK). Note: Values shown are the means ± SE (*n* = 4). Lowercase letters indicate significant differences at the $p < 0.05$ level between the control and N-treatment plots.

As expected, the rhizosphere soil $NH_4^+$-N and $NO_3^-$-N concentrations of all three species were generally higher in the N addition treatment than in the control, while the available P concentration and AP and NAG activities were unaffected by N addition in both rhizosphere and bulk soils (except for NAG activity, which significantly increased in the rhizosphere soil of *L. gracile* in response to the N addition treatment) (Table 2).

**Table 2.** Available nutrients and enzyme activities in rhizosphere and bulk soils of the three understory plant species in the Chinese fir plantation treated by nitrogen addition or in the control.

| Variables | Control | Nitrogen Addition | *T*-test |
|---|---|---|---|
| **Rhizosphere Soil** | | | |
| *Dicranopteris dichotoma* | | | |
| $NH_4^+$-N (mg kg$^{-1}$) | 16.03 ± 0.52 | 34.22 ± 0.89 | $p < 0.05$ |
| $NO_3^-$-N (mg kg$^{-1}$) | 1.68 ± 0.11 | 2.93 ± 0.35 | $p < 0.05$ |
| Available P (mg kg$^{-1}$) | 3.81 ± 0.19 | 3.57 ± 0.57 | ns |
| NAG activity (mmol g$^{-1}$ h$^{-1}$) | 30.5 ± 8.47 | 14.3 ± 3.45 | ns |
| AP activity (mmol g$^{-1}$ h$^{-1}$) | 358.3 ± 57.33 | 419.8 ± 39.29 | ns |
| *Lophatherum gracile* | | | |
| $NH_4^+$-N (mg kg$^{-1}$) | 9.95 ± 0.66 | 19.56 ± 0.23 | $p < 0.05$ |
| $NO_3^-$-N (mg kg$^{-1}$) | 1.85 ± 0.31 | 3.04 ± 0.85 | ns |
| Available P (mg kg$^{-1}$) | 3.82 ± 0.36 | 3.40 ± 0.15 | ns |
| NAG activity (mmol g$^{-1}$ h$^{-1}$) | 3 ± 2.38 | 20.8 ± 5.15 | $p < 0.05$ |
| AP activity (mmol g$^{-1}$ h$^{-1}$) | 231.5 ± 52.34 | 303.5 ± 32.75 | ns |

**Table 2.** *Cont.*

| Variables | Control | Nitrogen Addition | *T*-test |
|---|---|---|---|
| *Melastoma dodecandrum* | | | |
| $NH_4^+$-N (mg kg$^{-1}$) | 7.83 ± 0.71 | 12.49 ± 0.74 | $p < 0.05$ |
| $NO_3^-$-N (mg kg$^{-1}$) | 1.16 ± 0.06 | 2.21 ± 0.37 | $p < 0.05$ |
| Available P (mg kg$^{-1}$) | 3.02 ± 0.42 | 3.77 ± 0.27 | ns |
| NAG activity (mmol g$^{-1}$ h$^{-1}$) | 22.5 ± 4.575 | 28.5 ± 7.89 | ns |
| AP activity (mmol g$^{-1}$ h$^{-1}$) | 272 ± 15.44 | 356.3 ± 56.79 | ns |
| **Bulk Soil** | | | |
| *Dicranopteris dichotoma* | | | |
| $NH_4^+$-N (mg kg$^{-1}$) | 12.68 ± 3.09 | 31.99 ± 0.76 | $p < 0.05$ |
| $NO_3^-$-N (mg kg$^{-1}$) | 1.55 ± 0.17 | 2.74 ± 0.84 | ns |
| Available P (mg kg$^{-1}$) | 3.16 ± 0.14 | 3.46 ± 0.36 | ns |
| NAG activity (mmol g$^{-1}$ h$^{-1}$) | 26.01 ± 12.45 | 33.81 ± 3.04 | ns |
| AP activity (mmol g$^{-1}$ h$^{-1}$) | 344.82 ± 88.11 | 372.31 ± 29.19 | ns |
| *Lophatherum gracile* | | | |
| $NH_4^+$-N (mg kg$^{-1}$) | 9.66 ± 0.81 | 18.66 ± 1.20 | $p < 0.05$ |
| $NO_3^-$-N (mg kg$^{-1}$) | 1.39 ± 0.09 | 3.23 ± 0.26 | $p < 0.05$ |
| Available P (mg kg$^{-1}$) | 4.06 ± 0.45 | 3.06 ± 0.41 | ns |
| NAG activity (mmol g$^{-1}$ h$^{-1}$) | 12.80 ± 7.98 | 19.55 ± 6.88 | ns |
| AP activity (mmol g$^{-1}$ h$^{-1}$) | 237.8 ± 21.17 | 317.3 ± 20.07 | ns |
| *Melastoma dodecandrum* | | | |
| $NH_4^+$-N (mg kg$^{-1}$) | 8.18 ± 1.47 | 14.88 ± 0.52 | $p < 0.05$ |
| $NO_3^-$-N (mg kg$^{-1}$) | 0.92 ± 0.08 | 2.48 ± 0.37 | $p < 0.05$ |
| Available P (mg kg$^{-1}$) | 3.77 ± 0.27 | 3.54 ± 0.24 | ns |
| NAG activity (mmol g$^{-1}$ h$^{-1}$) | 13.34 ± 7.78 | 29.51 ± 10.51 | ns |
| AP activity (mmol g$^{-1}$ h$^{-1}$) | 361.3 ± 36.82 | 314.3 ± 19.26 | ns |

The values are the means ± SE (*n* = 4). Available P = Available phosphorus, NAG = N-acetyl-β-D-glucosaminidase, AP = acid phosphatase; ns = not significant at $p > 0.05$ level.

## 3.2. Nutrients and NSCs in Plant Tissues

Nitrogen addition, plant species, and plant tissue had significant interactive effects on the nutrients and NSC concentrations of the three understory plants (Figure 2). The average tissue N concentration increased, P and ST concentrations did not change, and average SS and NSC concentrations decreased in response to the N addition treatment (Figure 2). When analyzed by species and tissue type, N addition significantly increased the N concentration in both tissue types of all three species (Figure 2a); significantly increased the P concentration in the aboveground tissue of *M. dodecandrum* (Figure 2b); significantly decreased the ST concentration in the aboveground tissue of *M. dodecandrum* and belowground tissue of *D. dichotoma* (Figure 2d); significantly decreased the SS in the aboveground and belowground tissues of *M. dodecandrum* and the aboveground tissue of *D. dichotoma* (Figure 2c); and significantly decreased the NSC concentrations of *M. dodecandrum* while significantly increasing the NSC concentration in the aboveground tissue of *L. gracile* (Figure 2e).

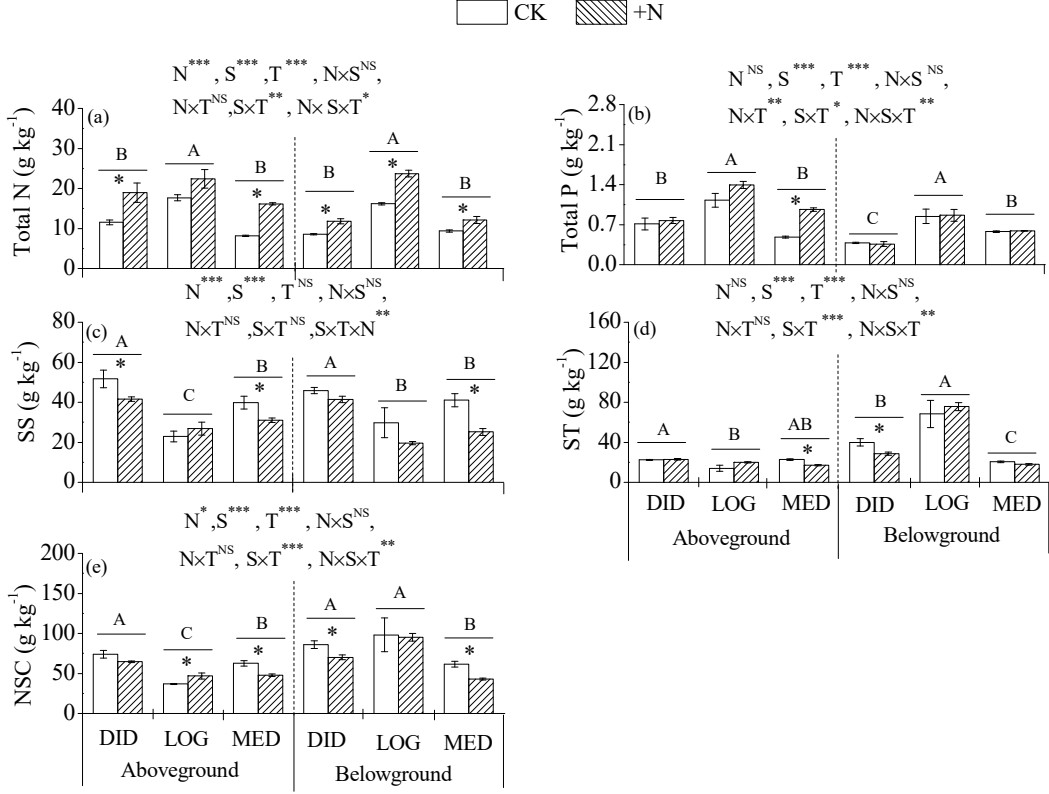

**Figure 2.** Total nitrogen (**a**), total phosphorus (**b**), soluble sugar (**c**), starch (**d**) and nonstructural ccarbohydrates (**e**) concentrations in aboveground and belowground tissues of the three understory plant species in the Chinese fir plantation treated by nitrogen addition (+N) or in the control (CK). Note: Values shown are the means ± SE (*n* = 4). The asterisks (*) indicate significant differences at the *p* < 0.05 level between the control and N-treatment plots within a single species. Different capital letters indicate significant differences (*p* < 0.05) among the three species within the same tissue based on Duncan's multiple range test. Total N = total nitrogen, Total P = total phosphorus, SS = soluble sugar, ST = starch, NSC = nonstructural carbohydrates; N = nitrogen addition, S = species, T = tissues. $^{NS}$, not significant; * *p* < 0.05; ** *p* < 0.01; *** *p* < 0.001. DID: *D. dichotoma*, LOG: *L. gracile*, MED: *M. dodecandrum*.

### 3.3. The Ratios of N, P, and NSCs in Plant Tissues

Similarly, N addition, plant species, and plant tissue had significant interactive effects on the ratios of N, P, and the NSCs of the major understory plants (Figure 3). Nitrogen addition did not affect the N/P ratio in the aboveground tissue of any species, but it significantly increased the belowground N/P ratio in all plant tissues (Figure 3a). For *M. dodecandrum*, both the NSC/N and NSC/P ratios significantly decreased in the aboveground and belowground tissues due to N addition; for *D. dichotoma*, only the NSC/N ratio in the aboveground tissue significantly deceased with N addition; and for *L. gracile*, both the NSC/N and NSC/P ratios were unaffected by N addition (Figure 3b,c).

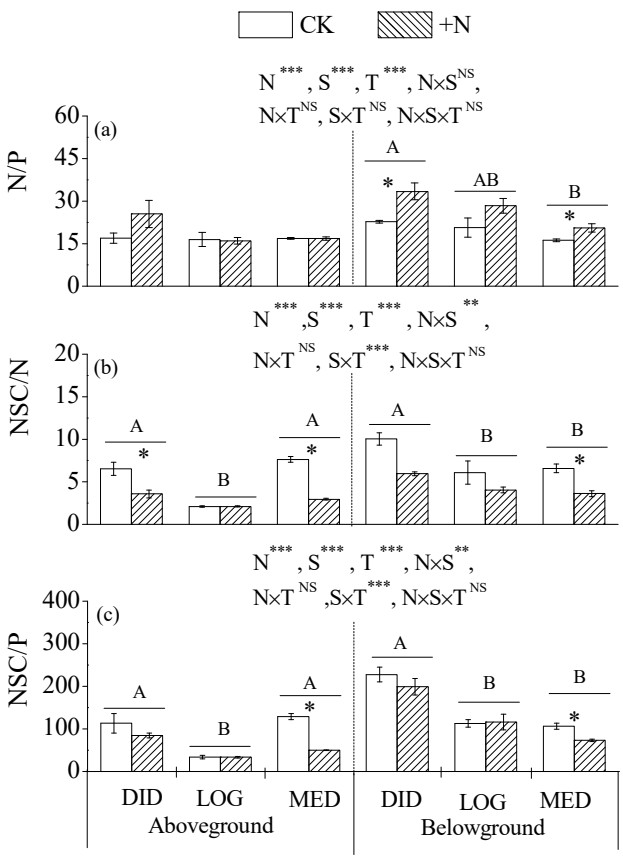

**Figure 3.** N/P (**a**), NSC/N (**b**), NSC/P (**c**) in aboveground and belowground tissues of the three understory plants in the Chinese fir forest plantation treated by nitrogen addition (+N) or in the control (CK). Note: Values shown are the means $\pm$ SE (*n* = 4). The asterisks (*) indicate significant differences at the $p < 0.05$ level between the control and N-treatment plots within a single species. Different capital letters indicate significant differences ($p < 0.05$) among the three species within the same tissue based on Duncan's multiple range test. N = nitrogen, P = phosphorus, NSC = nonstructural carbohydrates; N = nitrogen addition, S = species, T = tissues. $^{NS}$, not significant; * $p < 0.05$; ** $p < 0.01$; *** $p < 0.001$. DID: *D. dichotoma*, LOG: *L. gracile*, MED: *M. dodecandrum*.

### 3.4. Linkages between Plant Tissue Acquirement and Rhizosphere Soil Supply

The tissue N concentration was generally and positively correlated with rhizosphere soil $NH_4^+$-N and/or $NO_3^-$-N for each of the three understory plants (Table 3). The belowground tissue P concentration was significantly and positively correlated with rhizosphere soil available P for *D. dichotoma* and *L. gracile* but not for *M. dodecandrum*, while the aboveground tissue P concentration was significantly and positively correlated with rhizosphere soil $NO_3^-$-N for *L. gracile* and *M. dodecandrum* but not for *D. dichotoma* (Table 3). The belowground tissue NSC concentration was negatively correlated with rhizosphere soil $NH_4^+$-N for *M. dodecandrum* and *D. dichotoma* and positively correlated with rhizosphere soil available P for *L. gracile*, while the aboveground tissue NSC concentration was positively correlated with rhizosphere soil $NH_4^+$-N and $NO_3^-$-N for *L. gracile* and negatively correlated with rhizosphere soil $NH_4^+$-N for *M. dodecandrum* (Table 3). Likewise, the correlations among the ratios of N/P, NSC/N, NSC/P and soil available nutrients varied with plant species and tissues (Table 3).

**Table 3.** The correlation coefficients ($n = 8$) between plant tissue and rhizosphere soil nutrients for each of the three understory plant species in the Chinese fir plantation.

| Plant Variables | N | P | NSCs | N/P | NSCs/N | NSCs/P |
|---|---|---|---|---|---|---|
| **Belowground Tissue** | | | | | | |
| *Dicranopteris dichotoma* | | | | | | |
| $NH_4^+$-N | 0.88 ** | −0.15 ns | −0.74 * | 0.81 * | −0.89 ** | −0.41 ns |
| $NO_3^-$-N | 0.93 ** | 0.18 ns | −0.53 ns | 0.61 ns | −0.81 * | −0.53 ns |
| Available P | 0.29 ns | 0.93 ** | 0.22 ns | −0.53 ns | −0.04 ns | −0.66 ns |
| *Lophatherum gracile* | | | | | | |
| $NH_4^+$-N | 0.95 ** | 0.04 ns | −0.04 ns | 0.60 ns | −0.50 ns | 0.10 ns |
| $NO_3^-$-N | 0.64 ns | 0.77 * | 0.16 ns | −0.30 ns | −0.12 ns | −0.54 ns |
| Available P | −0.33 ns | 0.76 * | 0.73 * | 0.81 * | 0.01 ns | −0.84 ** |
| *Melastoma dodecandrum* | | | | | | |
| $NH_4^+$-N | 0.68 ns | 0.66 ns | −0.79 * | 0.60 ns | −0.78 * | −0.83 * |
| $NO_3^-$-N | 0.25 ns | 0.11 ns | −0.62 ns | 0.25 ns | −0.55 ns | −0.59 ns |
| Available P | 0.06 ns | 0.15 ns | −0.16 ns | −0.12 ns | −0.17 ns | 0.03 ns |
| **Aboveground Tissue** | | | | | | |
| *Dicranopteris dichotoma* | | | | | | |
| $NH_4^+$-N | 0.75 * | 0.16 ns | −0.57 ns | 0.57 ns | −0.76 * | −0.39 ns |
| $NO_3^-$-N | 0.90 ** | 0.16 ns | −0.60 ns | 0.71 * | −0.83 * | −0.42 ns |
| Available P | 0.51 ns | −0.16 ns | −0.15 ns | −0.32 ns | −0.02 ns | 0.56 ns |
| *Lophatherum gracile* | | | | | | |
| $NH_4^+$-N | 0.59 ns | 0.61 ns | 0.71 * | −0.10 ns | 0.11 ns | −0.06 ns |
| $NO_3^-$-N | 0.84 ** | 0.75 * | 0.86 ** | 0.03 ns | −0.31 ns | −0.06 ns |
| Available P | −0.20 ns | 0.37 ns | −0.14 ns | 0.33 ns | −0.55 ns | −0.53 ns |
| *Melastoma dodecandrum* | | | | | | |
| $NH_4^+$-N | 0.85 ** | 0.89 ** | −0.88 ** | −0.30 ns | −0.92 ** | −0.94 ** |
| $NO_3^-$-N | 0.77 * | 0.83 * | −0.45 ns | −0.38 ns | −0.70 ns | −0.72 * |
| Available P | 0.52 ns | 0.51 ns | −0.64 ns | −0.58 ns | −0.56 ns | 0.10 ns |

NSCs = nonstructural carbohydrates; ns, not significant, $p > 0.05$; * $p < 0.05$; ** $p < 0.01$.

## 4. Discussion

### 4.1. Links between Plant Tissue Acquirement and Rhizosphere Soil Supply

Similar to the results in many previous studies [3,30], soil available N, including $NH_4^+$-N and $NO_3^-$-N, increased with N addition in the form of $NH_4NO_3$. Consequently, in our study, the N concentrations in aboveground and belowground tissues were found to be higher in response to N addition for the three studied species. This result was consistent with those from previous studies in forest ecosystems [31–33]. Gurmesa found that plants could take up excessive N even in a N-saturated forest [34], which supported our results, as our plantation is located in a N-rich ecosystem within a region with extreme N deposition [35–37]. Thus, the higher soil N supply increased the N concentrations in aboveground and belowground tissues in response to N addition.

Surprisingly, the available P concentration and activities of both enzymes (NAG and AP) in rhizosphere and bulk soils of the three species generally did not change with N addition. Dong et al. (2015) also found the soil total P did not alter by N addition, while organic matter and total N increased due to three years' N addition of 100 kg hm$^{-2}$ year$^{-1}$ in the same site of this study [26]. These results indicated that the intensity and duration of the simulated N addition might still be within the buffer range of the soil P supply in the studied plantation [3]. Thus, unsurprisingly, the P concentration in plant tissues was generally unaffected by N addition in our study. However, the P concentration in aboveground tissue of *M. dodecandrum* did increase with N addition. In general, the effect of N addition on plant P may be altered via P supply, P uptake and P resorption [38–40]. In our study, soil available P and belowground tissue P showed minimal responses to N addition. Some studies

suggested that N addition decreased P uptake by reducing root biomass and inhibiting mycorrhiza growth [10,41]. Other studies found that N addition decreased the extraradical hyphae of Chinese fir mycorrhiza, but did not alter fine root biomass in a subtropical plantation forest [42]. Thus, we speculate that P resorption might be a potential mechanism increasing aboveground P concentration in *M. dodecandrum*. Compared with the other two species (a sun fern and a neutral grass), *M. dodecandrum* is a creeping shrub with traits characteristic of shade plants and is thus better suited to growing in an understory environment (Figure A1). To meet the requirements for growth, *M. dodecandrum* (the only studied species that did not exhibit decreased biomass with N addition) might improve P resorption to overcome the P insufficiency in soils and increase its aboveground P concentration to maintain N/P stoichiometric homeostasis.

As expected, the ratio of mineral N to available P in soils and the N/P ratio in belowground tissues generally increased with N addition, which partially supported our first hypothesis that N addition alters the balance between N and P in soils and understory plants. However, the N/P ratio in the aboveground tissue was not affected by N addition in any species, which seems counter to our first hypothesis of an imbalance between N and P in understory plants driven by N addition. The divergent patterns of the N/P ratio between aboveground and belowground tissues indicated that the understory plants might maintain N/P homeostasis in aboveground tissue to meet the requirement for leaf photosynthesis and plant growth [13,43], regardless of the alteration of the N/P ratio in plant belowground tissues due to N addition.

To achieve N/P homeostasis in aboveground tissue, the three understory species might use different mechanisms, such as the resorption in *M. dodecandrum* mentioned above, which should be further studied. The N/P ratio in plant tissue has been widely used as a diagnostic tool for evaluating nutrient limitation in terrestrial ecosystems [44,45]. Previous studies observed that the N/P ratio in tree leaves decreased with N deposition, and this indicated that N deposition might aggravate P limitation of plant growth in forest ecosystems [46]. In our study, the N/P ratio of aboveground tissue was generally less than 16, except for in *D. dichotoma* treated by N addition, and the N/P ratio in aboveground tissue was unchanged by N addition. These results suggested that N addition might not aggravate the P limitation of understory plant growth in this subtropical plantation.

### 4.2. Effects of N Addition on Understory Plant Growth and NSC Allocation

Understory plant growth might be altered by N addition through a direct effect of nutrient supply and an indirect effect of the overstory canopy via competition for resources such as light and water [3,14,31]. In our study, the biomass of two dominant understory species (*D. dichotoma* and *L. gracile*) significantly decreased with N addition. A recent study found that a moderate supply of N stimulated the understory vegetation growth in a tropical forest because N inputs satisfied plant demands for N [14]. However, some research suggested that understory vegetation productivity was mainly limited by light in a boreal forest, and N addition increased shading by the tree canopy (thus, less light was available to understory plants) [12]. Some other studies also found that plant growth was inhibited by N addition due to the aggravation of P limitation, with a mismatch in N and P stoichiometry [47,48].

Based on the responses of soil and plant nutrients to N addition discussed above, N addition likely did not aggravate the P limitation of understory plant growth in this subtropical plantation. Therefore, the observed decline in understory plant biomass may be most likely caused by increased shading by stimulated overstory tree growth due to N addition. Compared with overstory trees, understory plants are more easily limited by light [10]. Our data also indicated that N addition promoted the growth and litter production of Chinese fir (the canopy tree) (Figure 1) and thus could lead to a decrease in the light available to understory vegetation. The decrease in light availability might help explain the negative effect on understory vegetation growth in the N addition plots.

The NSCs in plant tissue are the products of photosynthesis and the main energy sources for plant growth [29]. Because NSCs can reflect plant response to light and nutrient availability [49,50], they

may provide an effective way to reveal the underlying mechanism of N addition effects on understory plants driven mainly by light or nutrient resources. The storage of NSCs is lower in shade than in sun environments because carbohydrate synthesis is often limited by lower light availability [29]. In our study, NSCs, especially SS, concentrations in understory plants significantly decreased with N addition. However, previous studies found that carbohydrate reserves increased with elevated N supply when N inputs satisfied plant demand for N and increased the photosynthetic capacity [16,17]. There are several potential reasons for the reduction in NSC concentration of understory plants in response to N addition in our study. First, N addition may have increased the plant growth and decreased the accumulation of NSCs because the exogenous N supply stimulated the synthesis of amino acids and amide compounds to suppress the accumulation of carbohydrates for protein synthesis when N was in excess [49]. Second, an excessive concentration of N in aboveground tissue may have resulted in inorganic N toxicity, which may have downregulated the photosynthetic capacity [14]. Third, the light available to the understory plants may have significantly decreased because the tree canopy cover increased under the elevated N supply [51,52]. Our study found that the soil available N was strongly correlated with the N concentration in the tissues of the three understory species, which indicated that the deposited N might not have been excessive or toxic. We also found negative correlations between soil $NH_4^+$-N and tissue NSCs in *D. dichotoma* and *M. dodecandrum*, but both soil $NH_4^+$-N and $NO_3^-$-N were positively correlated with aboveground tissue NSCs in *L. gracile*. Therefore, our results further indicated that the growth of understory species might be limited by light availability but not dominated by N (excess or toxicity).

In addition, the ratio of NSCs to nutrients reflects the relationship between the nutrients and the production of NSCs and their use efficiencies [53]. In our study, the ratios of NSCs to nutrients (NSC/N and NSC/P) in plant tissues generally decreased with N addition, implying that each unit of N and each unit of P produced fewer NSCs. The decrease in the NSC/N and NSC/P ratios in tissues also suggested that light availability may limit the photosynthetic rate and growth of understory plants in response to N addition [14,31].

Our results revealed that the three understory species were not consistent in their response to N addition, although they grow well in acid soils. *D. dichotoma* was a high light-demanding species, and *M. dodecandrum* was a shade tolerance species, but *L. gracile* was a duality species that can survive in low light and high light environment. Nitrogen addition has a direct influence on overstory vegetation that changes the light available reaching the understory. It is the reason that explains the different responses to N addition among the three understory species. First, decreased in the biomass of high light-demanding *D dichotoma* due to N addition is a consequence of lower light availability induced by the promoted growth of the Chinese fir tree canopy. This result is consistent with that of other research in that N fertilization has a significant influence on forest tree canopies, which can significantly reduce the light available to the understory plants [51,52]. Second, N addition promoted the growth of the Chinese fir tree canopy, leading to *L. gracile* being light limited as well as increasing the belowground nutrient accumulation and microbial activity of *L. gracile*. Compared with the sun and neutral plants (*D. dichotoma* and *L. gracile*), the shade plant (*M. dodecandrum*) showed a stronger capacity to synchronously increase N and P levels and decrease SS and ST pools in its tissues to maintain productivity and adapt to the shadier environment in the N addition treatment. In contrast, P and NSCs in the sun and neutral plants showed a weaker response to the N addition treatment, which led to a decrease in plant biomass with N addition. Furthermore, the correlations among soil-available nutrients, tissue nutrients, and NSCs differed among species, which further indicated species-specific mechanisms in response to N addition due to the differences in nutrient- and light-related traits among the three understory species.

## 5. Conclusions

Nitrogen addition did not lead to a mismatch in N and P stoichiometry in the aboveground tissues of understory plants, although N addition increased N availability in soils and plants. However, N addition decreased NSCs, especially SS, in two of the three studied species and the NSC/N

and NSC/P ratios of all three species. Meanwhile, the aboveground biomass of *D. dichotoma* and *L. gracile* significantly decreased after four years of simulated N addition, and the biomass of *M. dodecandrum* did not respond to N addition. These results suggested that N addition might inhibit the growth of understory plants through decreasing the nonstructural carbohydrates and light availability indirectly rather than by changing nutrients and N/P stoichiometry directly, although species-specific responses to N addition occurred in the Chinese fir plantation. The limitation of available light to understory species through the facilitation of N addition on overstory canopy growth may be the underlying mechanism, and thinning should therefore be used to improve understory vegetation biomass and other potential functions to mitigate the adverse effects caused by N deposition on understory plant species.

**Author Contributions:** F.C. was responsible for funding acquisition and resources. F.C. and G.G.W. conceptualized the study. F.W. performed the data curation and investigation. F.C. and H.W. participated in the design of the study. X.F. and W.B. supervised the experimental process. F.W. wrote the original draft. R.M. reviewed and edited the manuscript. All authors read and approved the final manuscript.

**Funding:** This research was funded by [the National Natural Science Foundation of China] grant numbers [31730014 and 31870427] and [Jiangxi Provincial Department of Science and Technology] grant numbers [20153BCB22008, 20165BCB19006, and 20181ACH80006]. The APC was funded by [2011 Collaborative Innovation Center of Jiangxi Typical Trees Cultivation and Utilization, Jiangxi Agricultural University].

**Acknowledgments:** We thank Xiu-Lan Zhang, Zhang-Min Li and Gao-Yang Wu for their field work and sample analysis.

**Conflicts of Interest:** The authors have declared that no competing interests exist.

## Appendix A

**Table A1.** General properties of the Chinese fir forest plantation before experimental treatment in 2011.

| Variables | Control | Nitrogen Addition | *T*-test |
|---|---|---|---|
| Soil | | | |
| Bulk density (g cm$^{-3}$) | $1.22 \pm 0.03$ | $1.25 \pm 0.03$ | ns |
| pH | $4.43 \pm 0.04$ | $4.40 \pm 0.08$ | ns |
| Organic carbon (g kg$^{-1}$) | $21.20 \pm 2.2$ | $22.41 \pm 1.62$ | ns |
| Total nitrogen (g kg$^{-1}$) | $1.29 \pm 0.13$ | $1.26 \pm 0.09$ | ns |
| Total phosphorus (g kg$^{-1}$) | $0.29 \pm 0.02$ | $0.31 \pm 0.03$ | ns |
| Stand | | | |
| Density (ha$^{-1}$) | $2250 \pm 45$ | $2150 \pm 62$ | ns |
| Average DBH (cm) | $12.2 \pm 0.2$ | $11.9 \pm 0.3$ | ns |
| Average height (m) | $8.5 \pm 0.2$ | $8.7 \pm 0.2$ | ns |

Data are cited from our previous study [3]; ns indicates not significant between the control and the nitrogen addition treatment at the $p < 0.05$ level. Values are the means $\pm$ SE ($n = 4$). DBH = diameter at breast height.

**Table A2.** Girth growth rates at breast height in the Chinese fir forest plantation in the nitrogen addition treatment and control from 2012 to 2015.

| Variables | Control | Nitrogen Addition | *T*-test |
|---|---|---|---|
| Girth growth rate | | | |
| 1-year (mm year$^{-1}$) | $10.89 \pm 0.69$ | $13.69 \pm 1.14$ | $p < 0.05$ |
| 2-year (mm year$^{-1}$) | $15.13 \pm 0.70$ | $18.64 \pm 1.18$ | $p < 0.05$ |
| 3-year (mm year$^{-1}$) | $9.50 \pm 0.57$ | $10.66 \pm 0.60$ | ns |
| 4-year (mm year$^{-1}$) | $13.72 \pm 1.15$ | $15.23 \pm 1.14$ | ns |
| Average (mm year$^{-1}$) | $12.31 \pm 0.61$ | $14.56 \pm 0.88$ | $p < 0.05$ |

ns indicates not significant between the control and the nitrogen addition treatment at the $p < 0.05$ level. Values are the means $\pm$ SE ($n = 4$).

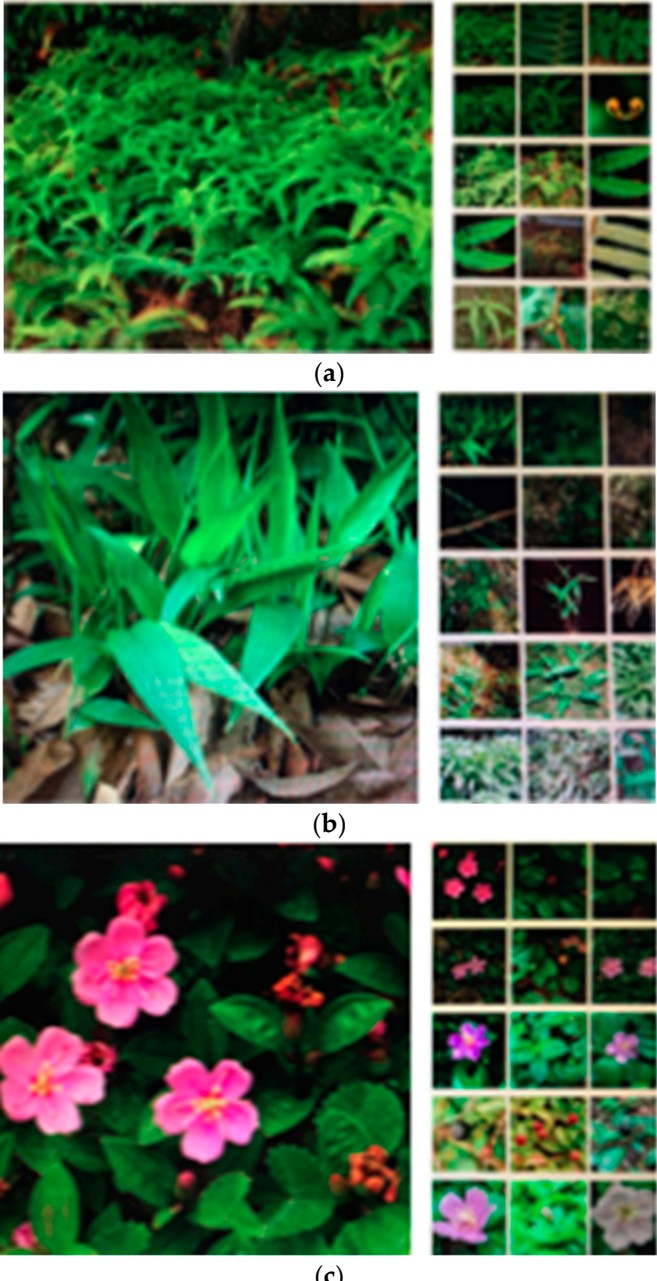

(**a**)

(**b**)

(**c**)

**Figure A1.** The morphological characteristics of the three species of understory plants in our study. (**a**) *Dicranopteris dichotoma* [54] (**b**) *Lophatherum gracile* [55] (**c**) *Melastoma dodecandrum* [56].

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
