# Peer review of "Effects of Experimental Nitrogen Addition on Nutrients and Nonstructural Carbohydrates of Dominant Understory Plants in a Chinese Fir Plantation"

_forests, doi:10.3390/f10020155_

Round 1
Reviewer 1 Report
This study shows results from a chronic N addition experiment conducted to simulate N deposition in a Chinese fir (Cunninghamia lanceolata) plantation of subtropical China.
Knowing the importance of atmospheric N in environmental responses and ecosystem services degradation, these type of experiments are really very useful and important. However the authors need to clarify the readers.
M&M - you should described better the study site: soil characteristics, (silt, loam and sand proportion), type of understory and diversity, maximum and minimum temperature of the site, as well as precipitation. Means are not enough. Also, it is important that you mentioned the average N deposition of the area, in order to understand why you have chosen those values.
How did you calculate understory richness? Number of total species?
D = s / √N
and this is different from species diversity, since this one takes into account the number of species present and the dominance or evenness of species in relation to one another. This information, particularly when we intend to show differences of treatment is much better information than only species richness.
You said that N application was added together with sand. Why did you need sand? You need to give a better information about your soil characteristics. Also it is important to know your care concerning the addition of fertiliser and precipitation because it would be completely different if you add fertiliser before a rainy season.
Did you find increment of bare soil due to N additions? This would be interesting to know, although you show that there is no differences in species richness. But since there is not clear the way you calculate it .. you should add a better explanation…
Also, it would be important to know what is the type of life-cycle of those chosen species. And this because we didn’t understand in what stage of development you make your tissue collections and analysis. Are no similar differences in growth stages of those plants?
Results
Surprisingly, the available P concentration and activities of both enzymes (NAG and AP) in rhizosphere and bulk soils of the three species generally did not change with N addition.
but above you add
Our research showed that NAG enzyme activity significantly increased, which may be directly associated with increased soil N availability after N addition.
This is an incongruence.
Discussion
M. dodecandrum is a creeping shrub with traits characteristic of shade plants and is thus better suited to growing in an understory environment (http://frps.eflora.cn). To meet the requirements for growth, M. dodecandrum (the only studied species that did not exhibit decreased biomass with N addition) might improve P resorption to overcome the P insufficiency in soils and increase its aboveground P concentration to maintain N/P stoichiometric homeostasis.
Did you look at roots? Do they form mycorrhiza? Have you any indication about this? Did you notice an increment of rhizospheric soil in this plant?
Our study found that the soil available N was strongly correlated with the N concentration in the tissues of the three understory species, which indicated that the deposited N might not have been excessive or toxic. We also found negative correlations between soil NH4+-N and tissue NSCs in D. dichotoma and M. dodecandrum, but both soil NH4+-N and NO3--N were positively correlated with belowground tissue NSCs in L. gracile. Therefore, our results further indicated that the growth of understory species might be limited by light availability but not 338 dominated by N (excess or toxicity).
This explanation seems to be too simple. Your argument that an increment of leaves on overstory is the explanation is not clear within your supplement data. You show only girth growth increment that was not repercute in DBH. So, how can you say - if you don’t show - an increment of shade
On the other hand, that explanation doesn’t fit with results from Fig. 2 since the major response is species dependent more than species treatment depend and with data from table 3, and described from lines 239 till 245.
In fact, the authors did not explain the possible differences due to different availability of N forms in the soil. Knowing that nitrate and ammonium lead to different metabolic processes and, thus, different functional plant adaptations (Dias et al 2011, 2015), plant diversity (Dias et al., 2014) and ecosystem responses (Dias et al., 2017) you should take into considerations all those factors before you make all your final statements.
Author Response
Reviewer 1
Open Review (x) I would not like to sign my review report
( ) I would like to sign my review report
English language and style ( ) Extensive editing of English language and style required
( ) Moderate English changes required
(x) English language and style are fine/minor spell check required
( ) I don't feel qualified to judge about the English language and style
Yes Can be improved Must be improved Not applicable
Does the introduction provide sufficient background and include all relevant references? (x) ( ) ( ) ( )
Is the research design appropriate? (x) ( ) ( ) ( )
Are the methods adequately described? ( ) ( ) (x) ( )
Are the results clearly presented? ( ) ( ) (x) ( )
Are the conclusions supported by the results? ( ) ( ) (x) ( )
Comments and Suggestions for Authors
Point 1: This study shows results from a chronic N addition experiment conducted to simulate N deposition in a Chinese fir (Cunninghamia lanceolata) plantation of subtropical China.
Knowing the importance of atmospheric N in environmental responses and ecosystem services degradation, these type of experiments are really very useful and important. However the authors need to clarify the readers.
Response 1: Thank you for your compliments. We revise the related contents according to your suggestions.
Point 2: M&M - you should described better the study site: soil characteristics, (silt, loam and sand proportion), type of understory and diversity, maximum and minimum temperature of the site, as well as precipitation. Means are not enough. Also, it is important that you mentioned the average N deposition of the area, in order to understand why you have chosen those values.
Response 2: Since the study site was introduced in our previous study (Chen et al. 2015 in Tree Physiology), we did not provide detailed information in original version of this study. Now, according to your suggestion, we add the related information as follow: The area is a typical red soil hilly region with a subtropical moist monsoon climate. The soil belongs to the typical Hapludult Ultisols with 68% sand and 15% clay (Xiong et al. 2015). The month average temperatures range from 6.5 °C in January to 29.7 °C in July, and a mean annual temperature is 18 °C. The annual precipitation ranges from 945 to 2144 mm with an average of 1500 mm (Kou et al. 2017). The area belongs to the core distribution of Chinese fir and the center of N deposition in China with 49 kg N ha−1 year−1 (Lü & Tian 2007; Chen et al. 2015). (line 94-99 in forest-435883 without trace change)
Chen F S, Niklas K J, Liu Y, et al. Nitrogen and phosphorus additions alter nutrient dynamics but not resorption efficiencies of Chinese fir leaves and twigs differing in age[J]. Tree physiology, 2015, 35(10): 1106-1117.
Kou L, Zhang X, Wang H, et al. Nitrogen additions inhibit nitrification in acidic soils in a subtropical pine plantation: effects of soil pH and compositional shifts in microbial groups[J]. Journal of Forestry Research, 1-10.
Lü C, Tian, H. Spatial and temporal patterns of nitrogen deposition in China: synthesis of observational data. Journal of Geophysical Research-Atmospheres, 2007, 112: 10-15
Xiong Y, Xu X, Zeng H, et al. Low Nitrogen Retention in Soil and Litter under Conditions without Plants in a Subtropical Pine Plantation[J]. Forests, 2015, 6(7): 2387-2404.
Point 3: How did you calculate understory richness? Number of total species?
D = s / √N
and this is different from species diversity, since this one takes into account the number of species present and the dominance or evenness of species in relation to one another. This information, particularly when we intend to show differences of treatment is much better information than only species richness.
Response 3: Yes, we use the number of each understory species in subplot as understory richness in this study. It is a potential flaw without considering the evenness of species. However, the dominance can be presented using the biomass of understory plants in our study. (line 112-113 in forest-435883 without trace change)
Point 4: You said that N application was added together with sand. Why did you need sand? You need to give a better information about your soil characteristics. Also it is important to know your care concerning the addition of fertiliser and precipitation because it would be completely different if you add fertiliser before a rainy season.
Response 4: Firstly, we mixed the fertilizer with sandy in order to ensure that a small amount of fertilizer can be evenly spread in the sample plot by the works. Secondly, we add the information of soil texture (68% sand, see the response above), and we consider that the added sand will not alter the soil physical characteristics since the proportion of accumulative sand added with continuous 10 years will not exceed 1% to the total sand amount within 0-20 cm soil depth. Thirdly, the fertilization weather is an important factor, thus fertilization generally choose the date without rain before 2 days and after 2 days combined with local weather forecast. (line 106-109 in forest-435883 without trace change)
Point 5: Did you find increment of bare soil due to N additions? This would be interesting to know, although you show that there is no differences in species richness. But since there is not clear the way you calculate it. you should add a better explanation…
Response 5: We’re sorry that we did not concern the issue mentioned by you. As response above, we use the number of each understory species as understory richness, and use the biomass of understory plants as the dominance of the species in our study.
Point 6: Also, it would be important to know what is the type of life-cycle of those chosen species. And this because we didn’t understand in what stage of development you make your tissue collections and analysis. Are no similar differences in growth stages of those plants?
Response 6: All those three species are perennial understory plants. We collected the samples of all these three understory species in maturation stage. (line 119-120 in forest-435883 without trace change)
Results
Point 7: Surprisingly, the available P concentration and activities of both enzymes (NAG and AP) in rhizosphere and bulk soils of the three species generally did not change with N addition.
but above you add
Our research showed that NAG enzyme activity significantly increased, which may be directly associated with increased soil N availability after N addition.
This is an incongruence.
Response 7: We checked our results as: the available P concentration and AP and NAG activities were unaffected by N addition in both rhizosphere and bulk soils (except for NAG activity, which significantly increased in the rhizosphere soil of L. gracile in response to the N addition treatment) (Table 2). Now, we deleted “Our research showed that NAG enzyme activity significantly increased, which may be directly associated with increased soil N availability after N addition.” in Discussion.
Discussion
Point 8: M. dodecandrum is a creeping shrub with traits characteristic of shade plants and is thus better suited to growing in an understory environment (http://frps.eflora.cn). To meet the requirements for growth, M. dodecandrum (the only studied species that did not exhibit decreased biomass with N addition) might improve P resorption to overcome the P insufficiency in soils and increase its aboveground P concentration to maintain N/P stoichiometric homeostasis.
Did you look at roots? Do they form mycorrhiza? Have you any indication about this? Did you notice an increment of rhizospheric soil in this plant?
Response 8: Thanks to provide a new perspective for this issue. We did not find the mycorrhiza for M. dedecandrum. But our colleagues noticed the mycorrhiza of Chinese fir in our study site. Li et al. (2019, Forest Ecology and Management) found that N addition decreased the extraradical hyphae, but did not alter fine root biomass. (line 272-275 in forest-435883 without trace change)
Li L, McCormack M L, Chen F, et al. Different responses of absorptive roots and arbuscular mycorrhizal fungi to fertilization provide diverse nutrient acquisition strategies in Chinese fir[J]. Forest Ecology and Management, 2019, 433: 64-72.
Point 9: Our study found that the soil available N was strongly correlated with the N concentration in the tissues of the three understory species, which indicated that the deposited N might not have been excessive or toxic. We also found negative correlations between soil NH4+-N and tissue NSCs in D. dichotoma and M. dodecandrum, but both soil NH4+-N and NO3--N were positively correlated with belowground tissue NSCs in L. gracile. Therefore, our results further indicated that the growth of understory species might be limited by light availability but not 338 dominated by N (excess or toxicity).
This explanation seems to be too simple. Your argument that an increment of leaves on overstory is the explanation is not clear within your supplement data. You show only girth growth increment that was not repercute in DBH. So, how can you say - if you don’t show - an increment of shade
Response 9: In order to draw the conclusion that the growth of understory species might be limited by light availability but not dominated by N (excess or toxicity), we need provide the evidence that light availability was altered by N addition. Unfortunately, it is difficult to directly obtain the light availability since the large variation both in spatial and temporal scales. In this study, we provide two indictors to indirectly show the light availability: girth growth increment at breast height and litterfall productivity. Our results show that the average girth growth rates of overstory trees (Chinese fir) within 4 years after N addition treatment and litterfall productivity in the fifth year significantly increased by 18.28% and 36.71%, respectively (Fig. 1).
Point 10: On the other hand, that explanation doesn’t fit with results from Fig. 2 since the major response is species dependent more than species treatment depend and with data from table 3, and described from lines 239 till 245.
Response 10: Firstly, we admit that the responses of understory plants to N addition varied with species. Fortunately, NSC especially SS concentrations in the three plants generally decreased with N addition. In addition, we discuss the potential mechanism for the unexpected results for L. gracile. Secondly, the statement that “but both soil NH4+-N and NO3--N were positively correlated with belowground tissue NSCs in L. gracile” shall be changed to “but both soil NH4+-N and NO3--N were positively correlated with aboveground tissue NSCs in L. gracile”.
Point 11: In fact, the authors did not explain the possible differences due to different availability of N forms in the soil. Knowing that nitrate and ammonium lead to different metabolic processes and, thus, different functional plant adaptations (Dias et al 2011, 2015), plant diversity (Dias et al., 2014) and ecosystem responses (Dias et al., 2017) you should take into considerations all those factors before you make all your final statements
Response 11: It is an important issue. In our study area, Kou et al. (2015, Plant Soil; 2017, Tree) explored the effect of N forms on forest ecosystem using the treatments of NH4Cl vs. NaNO3. In our study, we use NH4NO3 addition to replace N deposition, and do not consider the different metabolic processes in response to nitrate and ammonium. Fortunately, our study found that the soil available N was strongly correlated with the N concentration in the tissues of the three understory species, which indicated that the deposited N might not have been excessive or toxic.
Kou L, Guo D, Yang H, et al. Growth, morphological traits and mycorrhizal colonization of fine roots respond differently to nitrogen addition in a slash pine plantation in subtropical China[J]. Plant and Soil, 2015, 391(1-2): 207-218.
Kou L, Wang H, Gao W, et al. Nitrogen addition regulates tradeoff between root capture and foliar resorption of nitrogen and phosphorus in a subtropical pine plantation[J]. Trees, 2017, 31(1): 77-91.

Reviewer 2 Report
Understanding the way in which anthropogenic sources of nutrients influence forest ecosystem processes is of current concern. Atmospheric nitrogen downwind of urban areas can significantly alter plant production. Little attention has been given to the fate of understory plants under these conditions and this manuscript seeks to fill this gap in our knowledge. The manuscript is largely well written (but see below) and presents a good interpretation of the results found for eastern China. In some ways, however, the results are routine and do not necessarily pave new intellectual ground. It is, however, a clear description of the impacts of increased nitrogen availability on three understory plants that will be of interest to some.
Comments:
The manuscript is cleanly written, however, the Abstract had a number of awkward sentences. I suggest that it be revised for clarity.
Line 153: What was the homemade device? A dendrometer band?
Line 154: What was the size of the litter traps?
Results:
What was the effect of N addition on soil pH after the experimental treatments? Only the before treatment values are presented. I have experience acidification of soil from N addition.
Was any attempt made to directly measure light availability or canopy cover? This would add greatly to the manuscript as the results are largely the interaction of light and N availability.
Author Response
Open Review (x) I would not like to sign my review report
( ) I would like to sign my review report
English language and style ( ) Extensive editing of English language and style required
(x) Moderate English changes required
( ) English language and style are fine/minor spell check required
( ) I don't feel qualified to judge about the English language and style
Yes Can be improved Must be improved Not applicable
Does the introduction provide sufficient background and include all relevant references? (x) ( ) ( ) ( )
Is the research design appropriate? ( ) (x) ( ) ( )
Are the methods adequately described? ( ) ( ) (x) ( )
Are the results clearly presented? (x) ( ) ( ) ( )
Are the conclusions supported by the results? ( ) (x) ( ) ( )
Comments and Suggestions for Authors
Point 1: Understanding the way in which anthropogenic sources of nutrients influence forest ecosystem processes is of current concern. Atmospheric nitrogen downwind of urban areas can significantly alter plant production. Little attention has been given to the fate of understory plants under these conditions and this manuscript seeks to fill this gap in our knowledge. The manuscript is largely well written (but see below) and presents a good interpretation of the results found for eastern China. In some ways, however, the results are routine and do not necessarily pave new intellectual ground. It is, however, a clear description of the impacts of increased nitrogen availability on three understory plants that will be of interest to some.
Response 1: Thank you for your compliments
Comments:
Point 2: The manuscript is cleanly written, however, the Abstract had a number of awkward sentences. I suggest that it be revised for clarity.
Response 2: Done
Point 3: Line 153: What was the homemade device? A dendrometer band?
Response 3: The homemade device was vernier caliper (Guanglu SF 2000, made in China). (line 157-158 in forest-435883 without trace change)
Point 4: Line 154: What was the size of the litter traps?
Response 4:The size of the litter trap was 75 cm×75 cm.
Results:
Point 5: What was the effect of N addition on soil pH after the experimental treatments? Only the before treatment values are presented. I have experience acidification of soil from N addition.
Response 5: N addition decreased the soil pH after the experiment treatments which were verified in several researches (Dong et al. 2015; Ma et al. 2017).
Dong W Y, Zhang X Y, Liu X Y, et al. Responses of soil microbial communities and enzyme activities to nitrogen and phosphorus additions in Chinese fir plantations of subtropical China[J]. Biogeosciences, 2015, 12(18): 5537-5546.
Ma Z, Zhang X, Zhang C, et al. Accumulation of residual soil microbial carbon in Chinese fir plantation soils after nitrogen and phosphorus additions[J]. Journal of Forestry Research, 2017: 1-10.
Point 6: Was any attempt made to directly measure light availability or canopy cover? This would add greatly to the manuscript as the results are largely the interaction of light and N availability.
Response 6: As mentioned in the previous response, in order to draw the conclusion that the growth of understory species might be limited by light availability but not dominated by N (excess or toxicity), we need provide the evidence that light availability was altered by N addition. Unfortunately, it is difficult to directly obtain the light availability since the large variation both in spatial and temporal scales. In this study, we provide two indictors to indirectly show the light availability: girth growth increment at breast height and litterfall productivity. Our results show that the average girth growth rates of overstory trees (Chinese fir) within 4 years after N addition treatment and litterfall productivity in the fifth year significantly increased by 18.28% and 36.71%, respectively (Fig. 1).

Round 2
Reviewer 1 Report
The authors have improved the manuscript.
However they go on sustaining the explanation of light as the main driving force of those changes.
As I said since the beginning I realised that your data of girth (in mm increment) is nothing compared with your data in supplementary material of DBH that present no differences at all.
So, when you say
"Our data also indicated that N addition
365 promoted the growth and litter production of Chinese fir (the canopy tree) (Fig. 1) and thus led to a
366 decrease in the light available to understory vegetation. The decrease in light availability helped
367 explain the negative effect on understory vegetation growth in the N addition plots."
I go on saying that
It is too simple and too abusive saying that. You must say “may suggest….although…”.
Also, you always forget that more than N concentrations are the form of N plants are using and you have shown differences in all those rhizosphere. So, again, all your data can give ONLY some suggestions that you might study afterwards as a future proposal.
Author Response
Open Review
(x) I would not like to sign my review report
( ) I would like to sign my review report
English language and style
( ) Extensive editing of English language and style required
( ) Moderate English changes required
( ) English language and style are fine/minor spell check required
(x) I don't feel qualified to judge about the English language and style
Comments and Suggestions for Authors
The authors have improved the manuscript.
Point 1: However they go on sustaining the explanation of light as the main driving force of those changes. As I said since the beginning I realised that your data of girth (in mm increment) is nothing compared with your data in supplementary material of DBH that present no differences at all. So, when you say "Our data also indicated that N addition promoted the growth and litter production of Chinese fir (the canopy tree) (Fig. 1) and thus led to a decrease in the light available to understory vegetation. The decrease in light availability helped explain the negative effect on understory vegetation growth in the N addition plots." I go on saying that
It is too simple and too abusive saying that. You must say “may suggest….although…”. Also, you always forget that more than N concentrations are the form of N plants are using and you have shown differences in all those rhizosphere. So, again, all your data can give ONLY some suggestions that you might study afterwards as a future proposal.
Respond 1: We can understand your concern. Our data of Table A2 in supplementary material may be confused. In fact, the data showed that DBH of Chinese fir were not significantly different between N addition and control treatments in the third and fourth years after experiment, however, DBH significantly increased with N addition in the first and second years. Thus, the total DBH (the average girth growth rates at breast height in the past four years) significantly increased due to N addition. In this situation, we deduced that N addition promoted the growth of Chinese fir as shown in Fig. 1. Anyway, it is an indirect evidence to support that N addition led to a decrease in the light available to understory vegetation and the decrease in light availability helped explain the negative effect on understory vegetation growth in the N addition plots. Therefore, we change our conclusion as “N addition might inhibit the growth of understory plants by decreasing the nonstructural carbohydrates and light availability indirectly rather than by changing nutrients and N/P stoichiometry directly, although species-specific responses to N deposition occurred in the Chinese fir plantation.” (L39-42 and L397-400).

Reviewer 2 Report
Reviewer #1 has a number of substantial comments that need to be addressed. As regards their and my comments, I ask for the following clarifications in the manuscript:
1. The Abstract reads better but needs to include the fact that much of the response to N additions in understory plants were driven by decreased light availability (indirectly measured via changes in pine litterfall and bole growth).
2. When did the N treatments cease? In 2013 (as described in Dong et al. 2015) or in 2015 when the measurements for this study were taken? Please include this information in the manuscript.
3. Please cite the Dong et al. (2015) paper in the manuscript as a source of information on the treatments effects on soil properties. The pH effect was slight as were the other changes, but readers may want to investigate this further on their own.
4. It is still unclear how bole growth was measured. It now says a homemade device and a manufactured vernier caliper. What is the homemade device? I think the homemade device was a band of metal or plastic called a dendrometer, but this is not clear.
Author Response
Reviewer 2
Open Review
(x) I would not like to sign my review report
( ) I would like to sign my review report
English language and style
( ) Extensive editing of English language and style required
( ) Moderate English changes required
(x) English language and style are fine/minor spell check required
( ) I don't feel qualified to judge about the English language and style
Yes | Can be improved | Must be improved | Not applicable | |
Does the introduction provide sufficient background and include all relevant references? | (x) | ( ) | ( ) | ( ) |
Is the research design appropriate? | (x) | ( ) | ( ) | ( ) |
Are the methods adequately described? | ( ) | (x) | ( ) | ( ) |
Are the results clearly presented? | (x) | ( ) | ( ) | ( ) |
Are the conclusions supported by the results? | (x) | ( ) | ( ) | ( ) |
Comments and Suggestions for Authors
Reviewer #1 has a number of substantial comments that need to be addressed. As regards their and my comments, I ask for the following clarifications in the manuscript:
Point 1: The Abstract reads better but needs to include the fact that much of the response to N additions in understory plants were driven by decreased light availability (indirectly measured via changes in pine litterfall and bole growth).
Respond 1: We add the result in Abstract as “In contrast, N addition significantly increased the average girth growth rates and litterfall productivity of overstory trees by 18.28% and 36.71%, respectively.” (L34-36).
Point 2: When did the N treatments cease? In 2013 (as described in Dong et al. 2015) or in 2015 when the measurements for this study were taken? Please include this information in the manuscript.
Respond 2: In our study site, fertilization of N treatments have be developed since 2011. Soils were sampled twice in 2013 (two years after experiment) including July and November described in Dong et al. 2015. In our study, soils were sampled in August 2015 (the fourth year after experiment treatment). In addition, the soils for original properties were sampled before experiment of N addition in 2011.
Point 3: Please cite the Dong et al. (2015) paper in the manuscript as a source of information on the treatments effects on soil properties. The pH effect was slight as were the other changes, but readers may want to investigate this further on their own.
Respond 3: We cited the results of soil properties in Discussion (L280-282). We concerned that soil pH significantly decreased with N addition in Dong et al. (2015). In our study, we listed the data of pH for original soil. In both studies, soil P (available P in this study and total P in Dong et al. 2015) did not change due to N addition, and this is our key result to support our conclusion.
Dong W Y, Zhang X Y, Liu X Y, et al. Responses of soil microbial communities and enzyme activities to nitrogen and phosphorus additions in Chinese fir plantations of subtropical China[J]. Biogeosciences, 2015, 12(18): 5537-5546.
Point 4: It is still unclear how bole growth was measured. It now says a homemade device and a manufactured vernier caliper. What is the homemade device? I think the homemade device was a band of metal or plastic called a dendrometer, but this is not clear.
Respond 4: Now, we change to self-made dendrometer (including a sheet steel, a wire spring two steel nails and a digital caliper, see Chen et al. 2015).
